# Remote care for mental health: qualitative study with service users, carers and staff during the COVID-19 pandemic

Elisa Liberati ,[1] Natalie Richards ,[1] Jennie Parker ,[2] Janet Willars ,[3] David Scott ,[4] Nicola Boydell ,[5] Vanessa Pinfold ,[2] Graham Martin ,[1] Mary Dixon-Woods ,[1] Peter Jones [6]

► Prepublication history and additional online supplemental material for this paper are available online. To view these files, please visit the journal online ().

¹Department of Public Health and Primary Care, University of Cambridge, THIS Institute, Cambridge, UK
²McPin Foundation, London, UK
³Department of Health Sciences, University of Leicester, Leicester, UK
⁴Population Health and Genomics, University of Dundee, Dundee, UK
⁵Centre for Biomedicine Self and Society, Usher Institute, The University of Edinburgh, Edinburgh, UK
⁶Department of Psychiatry, Cambridge University, Cambridge, UK

**Correspondence to**
Professor Peter Jones;
pbj21@cam.ac.uk

## ABSTRACT

**Objectives** To explore the experiences of service users, carers and staff seeking or providing secondary mental health services during the COVID-19 pandemic.

**Design** Qualitative interview study, codesigned with mental health service users and carers.

**Methods** We conducted semistructured, telephone or online interviews with a purposively constructed sample; a lived experience researcher conducted and analysed interviews with service users. Analysis was based on the constant comparison method.

**Setting** National Health Service (NHS) secondary mental health services in England between June and August 2020.

**Participants** Of 65 participants, 20 had either accessed or needed to access English secondary mental healthcare during the pandemic; 10 were carers of people with mental health difficulties; 35 were members of staff working in NHS secondary mental health services during the pandemic.

**Results** Experiences of remote care were mixed. Some service users valued the convenience of remote methods in the context of maintaining contact with familiar clinicians. Most participants commented that a lack of non-verbal cues and the loss of a therapeutic 'safe space' challenged therapeutic relationship building, assessments and identification of deteriorating mental well-being. Some carers felt excluded from remote meetings and concerned that assessments were incomplete without their input. Like service users, remote methods posed challenges for clinicians who reported uncertainty about technical options and a lack of training. All groups expressed concern about intersectionality exacerbating inequalities and the exclusion of some service user groups if alternatives to remote care are lost.

**Conclusions** Though remote mental healthcare is likely to become increasingly widespread in secondary mental health services, our findings highlight the continued importance of a tailored, personal approach to decision making in this area. Further research should focus on which types of consultations best suit face-to-face interaction, and for whom and why, and which can be provided remotely and by which medium.

## Strengths and limitations of this study

► Strengths include its qualitative approach in speaking to a large sample of participants with varied mental health difficulties, carers and a diverse range of mental healthcare staff.
► Its novelty lies in a deep exploration of the views and experiences of remote mental healthcare during a pandemic.
► The methods are strengthened by the involvement of experts by experience and the use of peer research methods.
► The interviews were one-off conversations, so we could not explore change as the pandemic progressed and people may have become accustomed to remote care.
► The study used remote methods to comply with UK lockdown regulations; this will have excluded some groups without the ability to engage remotely.

## INTRODUCTION

Difficulties in mental health are very common; they bring long-term challenges for individuals, families, carers and society.[1] People with significant mental health needs may use secondary health services for specialised healthcare including acute in-patient services and community-based approaches such as early intervention, crisis resolution or specific therapeutic interventions for particular concerns. During the COVID-19 crisis in the UK and elsewhere, the number of people in need of mental healthcare increased. Besides those who suffered physically with COVID-19 itself, fear of infection, worry about those unwell and bereavement have been widespread, while measures such as lockdowns and other interventions to reduce transmission increased social isolation, loneliness and domestic strains; all create adverse conditions for mental health.[2–7] Yet, as need

increased, the capacity of mental healthcare provision was severely restricted due to distancing measures, extra hygiene precautions, abrupt changes to care pathways and reduced staff availability.[8] These changes saw provision and use of mental healthcare decrease[4 9] despite growing need and risk for service users.[10]

The COVID-19 pandemic forced the introduction of remote care across secondary mental health services in a matter of weeks. Many services switched to different forms of remote care as a way of increasing capacity and reducing face-to-face contact. Telehealth, where telephone and other technology-based methods are used to provide care,[11] has increased rapidly following years of inertia or slow growth.[12] Accompanying this rise has been a rapid expansion in research, evaluation, guidance and commentary on remote care in a range of fields. This body of literature to date has offered largely positive accounts of the potential of telehealth during, and beyond, the pandemic,[13] leading some to suggest that it may, at least in part, replace in-person interaction as a mode of healthcare delivery for many service users.[14] The bulk of research to date, however, has taken place in primary care and/or physical health. Of 543 papers identified by Doraiswamy *et al* in their rapid scoping review of articles relating to telehealth during COVID-19, for example, only 42 (7.7%) focused on psychiatry and related disciplines.[13]

Both the advantages and the challenges of remote care delivery in physical healthcare settings may differ substantially from those faced in secondary mental healthcare. The use of remote care in mental health has invited debate for decades.[15] Though some have highlighted the potential of telehealth in addressing mental health difficulties during and beyond the pandemic,[16] remote care may also have important downsides; some, such as difficulties with access to high-speed internet connections required to support videoconferencing, are more readily anticipated than others—such as the multiplicity of online platforms for mental healthcare and uncertain quality control.

Rigorous qualitative studies examining the experiences and needs of service users, carers and clinical staff involved in remote mental healthcare are needed to understand its impacts, and guide short and long-term changes to services. This is not only to mitigate problems but also to take advantage of opportunities to address long-standing concerns about access that have been exposed by the pandemic. In this article, we respond to this challenge. We report a large, interview-based study involving people with direct experience of seeking (including carers) and providing mental healthcare in England during the first wave of the COVID-19 pandemic.

## METHODS

The study was designed and developed with six experts by experience (three service users and three carers) and a peer researcher from the McPin Foundation, a mental health research charity.

Between June and August 2020, we undertook a qualitative study involving remote interviews with three groups: first, adults with mental health difficulties under the care of secondary mental health services who either accessed support, including inpatient and community mental health services, during the pandemic, or needed services but did not access them. We recruited people with experience of mental health services in England only. We did not include individuals seeking to access mental healthcare for the first time through their general practitioner or staff working in primary mental health services, such as general practitioner practices, community pharmacists and improving access to psychological therapy services. Second, we interviewed carers of people who accessed, or needed to access, secondary mental healthcare during COVID-19. Our third group of participants were drawn from those working in National Health Service (NHS) secondary (inpatient and community) mental health services—particularly those likely to be involved in critical and time-sensitive decisions.

Across participant groups, our recruitment strategy was informed by efforts to maximise diversity using a purposive sampling[17] to access a variety of experiences related to our research questions. We did not seek to achieve statistical representation of the population under study, but instead to reflect diversity. As data collection and analysis progressed in parallel, the size of the sample was adapted to the variety of experiences captured, in line with the principle of information power.[18]

We recruited individuals using online network-based approaches: some participants engaged in response to information circulated through dedicated networks, while others became involved as a result of colleagues or friends alerting them to the study (a technique known as snowball sampling). Multiple channels were used to publicise the study, including the networks of The Healthcare Improvement Studies (THIS) Institute and the McPin Foundation, Health Education England's Heads of Schools of Psychiatry, National Institute for Health Research (NIHR) Applied Research Collaborations, specialty clinical networks and mental health charities including Rethink Mental Illness.

Information about the study was circulated via email. People who wished to take part in the study completed an online expression of interest form, which included questions about their ethnicity, the gender they identified with, and the first half of their postcode. In line with our sampling approach, we reviewed responses to ensure diversity of experience, geography, minority background and gender identity. For staff participants, we also prioritised diversification of staff roles and levels of seniority.

To comply with lockdown restrictions, all interviews had to be conducted remotely. Potential participants were contacted by the researchers via telephone or email, depending on their preferred contact method. Eligible potential participants were provided with a link

to register via Thiscovery, a secure citizen-science platform developed by THIS Institute according to level AA Web Content Accessibility Guidelines that assure accessibility standards. Once registered on Thiscovery, potential participants were given further information and invited to complete the informed consent form. They were then able to schedule an interview, with the choice of videoconference (on Thiscovery) or a telephone call.

EL, NR, JW and JP conducted the interviews; JP is a lived experience researcher. The interview topic guide (online supplemental file) covered a range of themes we sought to explore; the guide was deliberately non-directive to allow participants to discuss areas they perceive as relevant such as feelings of abandonment for service users or moral injury for staff. Participants who opted for videoconference interviews had the option to turn their camera off. In both cases (videoconference or telephone call) only the audio was recorded. Interviews lasted between 22 min and 95 min. Service users and carers were compensated £25 for taking part in an interview.

Interview audio files were securely transferred to a third-party transcription service subject to the University of Cambridge data protection regulations. Anonymised service user, carer and staff interviews were analysed separately. Analysis of anonymised interview transcripts was based on the constant comparative method.[19] The coding scheme was developed based on a subset of initial interviews. The initial codes were revised, expanded and collapsed as analysis progressed, and through whole team discussions. Codes were then were organised into categories in a thematised coding scheme. Data were processed using NVIVO software by five coders (four females and one male, DS). JP and NB analysed service user interviews, EL analysed carer interviews, DS, NB and NR independently analysed staff interviews.

During the process of write up and dissemination, some interview excerpts were edited further to protect the identity of participants. We followed the Standards for Reporting Qualitative Research recommendations.[20]

All participants were provided with information about the study and gave consent.

## Patient and public involvement

We consulted a panel of mental health service users and carers convened by the McPin Foundation. Members helped us shape the research questions, methods and risk management plans for the study. We also gathered the views of these experts by experience on the study materials, including the research protocol and our participant-facing documents. We shared the study documents with these experts-by-experience requesting their comments and recommendations. We held a series of online meetings to give participants a further opportunity to share their views and suggestions for improvement, providing individual online meetings or a telephone call according to preference. Experts by experience provided advice on how best to include carers in the study and suggestions for specific networks for recruitment. They also guided

how best to compensate participants for their time, as well as emphasising the need for an accessible summary of the research to be made widely available.

Using peer-research methods,[21] a researcher from the McPin Foundation carried out and analysed the service user interviews, drawing on her own experiences of accessing secondary mental healthcare both before and during the study period.

## RESULTS

In total 69 people took part in the study (table 1). We interviewed 24 people with mental health difficulties under the care of secondary mental health services (19 by telephone, 5 by video). Of these, four interviews were excluded as they did not meet the eligibility criteria, so analysis was based on 20 interviews. We interviewed 10 individuals who cared for people with mental health difficulties (eight by telephone, two by video). We also interviewed 35 point of care staff (21 by telephone, 14 by video), including psychiatrists (trainees and consultants), care coordinators, mental health nurses, clinical psychologists and psychotherapists; some had managerial as well as clinical roles.

Most service users and carers who expressed interest in the study completed an interview. However, a large proportion of staff who expressed interest did not book an interview, likely because of pressures associated with the pandemic and job role moves. Reasonable diversity of participants was achieved (table 1), in line with our sampling strategy.

Participants reported that a widespread switch to remote care for secondary mental healthcare occurred in response to the pandemic. They described an evolving context where telephone was initially the most widely used technology (owing to its perceived wide accessibility), giving way, towards the end of our recruitment period (August 2020), to increased use of video-supported platforms. These platforms were introduced as issues about what was allowed by information governance policies began to be resolved, and as familiarity with the technology grew. In what follows, we report the accounts of participants in relation to: service user choices about remote care; the embodied dimensions of therapeutic encounters; remote assessments and identifying risks; and inequities in access.

### Service user choices about remote care

Service users had mixed experiences of remote care. Most reported that it was adequate or tolerable, but that face to face was much better. Where people did have positive experiences, they tended to be reported by service users who had a prior relationship with a clinician.

> [The psychiatrist] was really great across the phone…
> I was quite worried that the phone appointment was going to be terrible because I've always had it in

**Table 1** Participants' demographic information

| | Service users | Carers | Staff |
|---|---|---|---|
| **No of people who expressed interest in the study** | 60 | 18 | 142 |
| **No of people invited to interview** | 30 | 18 | 91 |
| **No of eligible interviews** | 20 | 10 | 35 |
| **Gender identity** | ► 8 women<br>► 7 men<br>► 2 non-binary<br>► 3 people did not provide this information | ► 6 women<br>► 2 men<br>► 2 people did not provide this information | ► 19 women<br>► 11 men<br>► 5 people did not provide this information |
| **Ethnicity** | ► 8 White<br>► 3 Black<br>► 2 Asian<br>► 4 Mixed ethnicity<br>► 3 people did not provide this information | ► 7 White<br>► 1 Asian<br>► 2 people did not provide this information | ► 24 White<br>► 3 Asian<br>► 2 Mixed ethnicity<br>► 1 from 'any other ethnic group'<br>► 5 people did not provide this information |
| **Region** | ► 1 North West<br>► 5 East Midlands<br>► 6 Greater London<br>► 2 East of England<br>► 1 South East<br>► 2 South West<br>► 3 people did not provide this information | ► 1 West Midlands<br>► 4 East of England<br>► 2 South East<br>► 1 South West<br>► two people did not provide this information | ► 4 North East<br>► 7 North West<br>► 2 East Midlands<br>► 5 West Midlands<br>► 4 Greater London<br>► 2 East of England<br>► 3 South East<br>► 3 South West<br>► 5 people did not provide this information |
| **Additional information** | Services accessed or contacted:<br>► Acute hospital wards<br>► Community Mental Health Teams (CMHT)<br>► Crisis Teams<br>► Specialist services (self-harm, personality disorder) | | ► 17 Psychiatrists (13 trainees and four consultants)<br>► 10 Mental health nurses (including care coordinators, matrons, non-clinical prescribers)<br>► 8 Clinical psychologists (including Cognitive Behavioural Therapists and systemic family therapists)<br>Services covered:<br>► Community Mental Health Teams<br>► Early intervention for psychosis<br>► Crisis Teams<br>► Acute hospital wards<br>► Secure Forensic services |

person, but he was really good. It was almost like he was in the same room as me. (Service user)

Several reported that trying to build a therapeutic relationship remotely with a member of staff who they did not know was problematic, and that they preferred being able to build on pre-established rapport.

I suppose if the AttendAnywhere works… actually, it doesn't take as long out of your day. It takes exactly an hour rather than travelling to the appointment. So, it's an option. But I still feel that the face-to-face has it for me. (Service user)

Service users often expressed dissatisfaction with the modality of remote care offered, particularly at the beginning of the pandemic when only telephone was mostly available. They reported feeling frustrated by the lack of choice they had in how to stay in contact with services, an

experience that was also shared by staff during the early stages of the pandemic.

It's like, oh, Zoom's not allowed because of privacy, confidentiality, then they tried to launch something and then it was rubbish. It took them like six or seven weeks and they still don't have their act together. Whereas I've heard [other] people… and they said that they can do it online so why is that [this Trust] can do it and you can't get your act together… (Service user)

Some service users reported that the use of remote methods made it more likely that appointment times would be altered or not followed through, leading to uncertainty and frustration.

I can't deal with uncertainty and inconsistency and I think they literally did…oh, we'll call at 3:00, and

then they'll call at 4:00 and it was, yes, we can, no, we can't. Then in the end they discharged me with a letter and then…it was in a way, you know when they discharge you as well, you don't know whether to say, yes, that's a good idea or no because you're worried that actually tomorrow I might have a really bad day (Service user)

I think it's a very poor service that make an agreement they'll ring somebody every week, and then suddenly to fall off a cliff like that, I think it's a really bad service, I'm sorry. I think someone should handover and say, well, I've got a vulnerable client that's actually been working with the [service name] as well, and then just to be dumped like that for nearly seven weeks, I'm sorry I think it's really poor. (Service user)

Some service users and carers chose not to receive remote care, particularly for psychological interventions, because they believed that face-to-face consultations would be reintroduced soon. Others did not feel that remote consultations would be as helpful or meaningful as face-to-face ones; they made clear that remote care was their only option during the pandemic, and was not a choice they would otherwise have made:

I thought it would be doubly frustrating… doing everything on the telephone and [that you'd] end up with endless telephone conversations that got nowhere. (Service user)

I certainly feel that I haven't made the progress with the psychologist… I was getting a lot more out of the appointments when they were face-to-face, I think. I don't come away feeling as if I've really taken on board what he's said to me in the same way. (Service user)

Staff expressed concerns that breaks in therapy might negatively affect service users' and families' mental health during the pandemic. They recognised the challenges in providing remote care and understood service users' choices not to engage with it, some offering face to face when allowed:

I did have quite a few drop out… [Some] chose not to continue because they didn't want to do remote working… Some people had an idea that [the pandemic] would be short lived… That was a concern for me because we were in the middle of therapy and I'm struggling now to get them back. (Family therapist)

I think that people have missed face-to-face contact… I've rung and said 'I'm going to go back to some places for contact… Do you want to see me, or do you want to just do it over the phone?' And everybody has said 'Yeah, would really love to see you.' (Care co-ordinator)

Some staff shared the view of many service users that remote care was more suitable for maintaining continuity in existing relationships than for meeting people who

were newly referred. Conversely, some clinical psychologists and psychotherapists reported that remote care was more likely to be accepted by service users who had not experienced face to face therapy before, because they did not endure the 'shock' of a change in access mode.

### The embodied dimensions of therapeutic encounters

The move to remote care helped to maintain some form of connection between service users and services during the pandemic. However, participants reported that remote care changed the character and depth of clinical encounters and, in turn, their relational quality. Uniting the accounts of service users, carers and staff was the loss of the embodied dimension of therapeutic interactions, including the physical space where these used to take place. Many described remote consultations as 'not the same', noting that even where they were able to see someone's facial expressions (in online consultations), not being together in a room meant that building and maintaining a connection was problematic. For example, when eye contact was mediated by a camera, it hampered the reciprocity normally experienced in face-to-face interactions, as did the emphasis on verbal communication over body language. These factors compounded many service users feeling socially isolated during the pandemic, while for others, the loss of a particular, even sacred, therapeutic setting undermined the quality of care:

[Remote consultations with psychiatrist] just felt more perfunctory, somehow, and I felt less like opening up about stuff. It tends to be quick discussions about my medication and that's it… It felt like there was a barrier and I was just less inclined to open up. (Service user)

You can't have simultaneous eye contact with someone. When you look at the camera, they have the experience that you're looking in their eyes but you don't have that experience because you're looking at the camera… It just creates a really weird asynchrony so it's just not like being in the room with the person… I just find that quite disconcerting. (Clinical psychologist)

Service users reported not feeling able to make full use of therapeutic interactions in an environment they shared with other people (commonly, family members). Others felt that video consultations were 'invading' their own private space:

In our home, we have three generations of families who live in a two-bedroom house. So, picture eight people in a two-bedroom house—you know, it's quite hard. (Service user)

There's something that… has that sacredness about the [consultation] room. (Service user)

The loss of the journey to and from services was another aspect of remote care that required adjustment. Some service users felt they were deprived of the opportunity to

process the content of the sessions and gently transition back to their home lives.

> After the phone call, he said 'Bye' and he put the phone down. I was just like 'Oh.' It just felt slightly surreal. (Service user)

> The trip… would at least get me into the headspace of 'Okay, I'm going into therapy.' And then, leaving, I would then get on a bus and just chill and let it sink. (Service user)

Similarly, some staff members reported less opportunity to reflect and process after consultations due to back-to-back appointments. Several therapists also emphasised the importance of therapy taking place in a safe and bounded physical space (the therapeutic setting). They saw remote consultations, often being delivered from their homes as removing a boundary between private life and therapeutic work, compromising the safety of these consultations. They found some types of therapeutic work, such as trauma processing, to be so ill-suited to remote care that they paused it.

> And I don't have any car journeys anymore. I really appreciated time in the car because it gave me time to reflect and time to process and time for myself. And in an office, home office with Microsoft Teams, you get a lot of appointments just back-to-back. (Family therapist)

> I had a girl… who started online, and, probably, about six sessions in, she got quite emotive… and then she suddenly shut down, and I was like 'What's that about?' And she said, 'Well, my mum's next door and I can't do that. This isn't safe.' (CBT therapist)

### Remote assessments and identifying risks
A particular area of concern for all participants related to the effectiveness of remote forms of care in conducting assessments and identifying risks. Service users reported that lack of face-to-face contact made it more challenging for staff to identify—and help them recognise themselves—signs that their mental health was changing. This was important when, for example, the nature of their mental health difficulties meant they could transition rapidly from depression to mania, without being able to understand that this was happening and that a crisis requiring inpatient admission might follow. In interviews, service users and carers described video consultations as going some way towards addressing these issues compared with telephone calls, but mostly they saw them as a poor substitute for face-to-face contact:

> In the run-up to my becoming very unwell I didn't see them face to face, and I think that I was becoming (I have bipolar) …. increasingly hypermanic [sic]. But, because I was still functioning at my job, I didn't really recognise it. (Service user)

> Before COVID, probably in face-to-face meetings there [was] more of an opportunity to observe body

language and assess mood from the physical presence of somebody that you're sitting with. I don't think that can be really captured over the phone. (Carer)

Some carers felt that the shift to remote care exacerbated a pre-pandemic problem of their being excluded from assessments, such that risk may not be assessed properly. One carer felt that the person they supported did not find remote consultations beneficial and tried to end the interactions quickly. Because the carer was not included in the consultations, some issues were not recognised, recorded or addressed:

> My [family member] just wants people off her back… [She] will say 'I'm fine' until she's blue in the face. It's much easier to do that on a half-hour phone call than it is possibly sitting face-to-face with somebody for an hour, who might actually be able to read further into things. (Carer)

Staff tended to agree on the inherent limitations of remote consultations for conducting assessments and identifying mental health risks—especially in the context of COVID-19, where service users presenting to community services were more acutely unwell. It appeared that specific aspects of assessments were particularly compromised by remote contact. For example, staff and carers said remote consultations made it difficult (or, in the case of telephone consultations, impossible) to pick up important non-verbal cues, such as body language and levels of tension. These difficulties were exacerbated for service users who struggled to verbally communicate how they were feeling. Acknowledging the limitations of remote consultations for risk assessment, some services used telephone consultations to evaluate the need for face-to-face contact. Yet sometimes this introduced other problems: delaying fuller assessments, for example:

> What a lot of doctors did was… a short assessment on the phone just to basically check for risks: it was more of an in-depth triage than a proper assessment. And then they would say, 'It is urgent that I see this person'. And they would do face-to-face assessment. I think what it did was just slow down the process from the point of referral to the point of deciding that we would take the person on to the caseload. (Care co-ordinator)

Other factors that limited the depth and appropriateness of remote assessments included an inability to evaluate service users' home circumstances. Staff had to rely much more on service users' self-report of their mental health without, in addition, picking-up subtle, non-verbal clues that are available during an interview. Staff acknowledged the importance of carers, family members or friends being physically present at the point of assessment, reporting that service users who did not have this support (eg, those living in care homes during the pandemic) may have been particularly disadvantaged. Remote working and lockdown restrictions also meant

that service users in compromising home life situations, such as those facing domestic abuse, may not have felt comfortable disclosing how they were coping:

> Domestic violence assessments—usually we would go out to the home and try and get a sense of how things were, whereas obviously on the phone you're only getting one side of the story. (Senior clinical psychologist)

> Our team would [normally] go to patients' homes and check their weight… but because of COVID we had to ask patients to do that. But our patients with anorexia, they have a tendency to falsify their weight. So, in that way, it would be difficult to give them quality care. (Trainee psychiatrist)

### Inequities in access

Participants across the service user, carer and staff groups reported multiple concerns about the potential for remote care to have disproportionately negative effects for some groups of service users and carers through intersectionality, thus creating or further amplifying inequities. These groups included those with hearing difficulties and communication impairments; people for whom interacting through a screen or a telephone was particularly challenging (eg, because of anxiety or autism); people who were unfamiliar with or reluctant to use technology, including some older people and those experiencing paranoia or delusions about technology; people who could not afford devices or internet access; and people who needed support with the English language, often impacting on whole communities (table 2).

Some participants emphasised the need to identify and address the inequalities created by new, as well as traditional, ways of providing support. For example, in evaluating the appropriateness of remote care, participants suggested that it is important to account for remote consultations that are missed or cancelled as well as those that do take place, to avoid selection bias:

> I'm just very fearful that [services] will take from this 'Oh, we can do it all on Zoom, we can do it all online, we can do it all on the phone. We don't need to actually see people.'… And that would be a very negative [thing] for an awful lot of people…. but those people would vanish quite rapidly. And then… the people who would be left would be those who are comfortable with that. And [services] would be able to say, 'Oh, look, it's working fine for these people.' Well, who have you lost on the way? (Carer)

| Table 2 | Groups identified as particularly at risk of disadvantage from remote care |
|---|---|
| **Groups at risk of disadvantage from remote care** | **Example quotations** |
| **Individuals with sensory (eg, hearing) difficulties and communication impairments**. | We work with over 65s, so we know that a lot of those are people who have hearing difficulties and it can be incredibly difficult for them to get the same out of the telephone session as they do in person. (Senior clinical psychologist)<br>I was working with a family that the [family member] had a stroke and can only write, and the family do a lot of her communicating in the family sessions. And they said there's just no way we can do that online, it's going to all… it's just going to stress her out completely. So people who have got additional needs that don't get met by the online platform. (Family therapist) |
| **Individuals for whom interacting through a screen or via telephone may be particularly challenging**. | I felt really not looked after in the community, the way they were proposing to help us was just calling, which is not very adequate for me because I have Asperger's and I really need something physical. (Service user)<br>I assessed a person with autism, and it was challenging. 'Cause I think if you struggle with human interaction in person, you probably struggle even more online. (Trainee psychiatrist) |
| **People who may be unfamiliar with, or unwilling to use, technology ('digital exclusion'), including older adults and individuals experiencing phobias or delusions regarding IT**. | Ours is an older population generally speaking, and not to stereotype, but a lot of the older population are not technology savvy, a lot don't have smart phones, a lot don't have iPads or computers. So, we haven't video called most of them, or a lot of them. It tends to be that we're just making lots and lots of telephone calls. (Care coordinator)<br>[Family member] won't Zoom. Part of his schizophrenia is he can't look at live television so, Zoom, he couldn't do. This is him personally I'm talking about rather than in general. But (…) I get to know about a lot of other cases… and we are unanimous(it's)not just schizophrenia (…)… Face to face is invaluable. (Carer)<br>There's one person who I'm still in contact with over the phone but she's phobic about technology, partly due to a previous trauma issues. (Clinical Psychologist) |
| **People who may not have access to technology or telephone/ internet contracts (including those without a secure accommodation)** | I have people who don't have internet access, don't have mobile phones, so if I don't go to their house, I'm not going to engage with them. (Care coordinator)<br>Well, I guess people in social economic kind of situations where they don't allow them to have the devices, so families that don't have devices or Wi-Fi, that might be more difficult. (Family therapist) |
| **Non-native English speakers (and those needing a language interpreter) for whom relaying only on verbal communication may constitute an obstacle to mutual understanding**. | But [remote access] is limited, it relies on people having good English, whereas we have quite a high Turkish population here. (Trainee psychiatrist)<br>A lot of our patients, because of the demographic, English may not be their first language or they may have an accent if they've grown up abroad, which can add to the difficulties in understanding people on the phone. (Trainee psychiatrist) |

I hope it doesn't go the other way and we end up cutting the face-to-face services too much. I do fear that in a way. I think the technological things would be the main positive change. I just hope that it doesn't come at the expense of saying, oh well, the face-to-face doesn't matter, or you can cut that, which in the context of older people is especially important. (Trainee psychiatrist)

Conversely, remote care appeared to have some role in addressing some inequalities associated with face to face only. For example, during the pandemic, it enabled people who were shielding or particularly vulnerable to the virus to continue accessing services, reducing transport costs and logistical difficulties for service users and carers, and benefitted people with physical impairments who were challenged by having to travel. That said, it did not benefit all.

For a long time, service users asked us 'Do we have to come all the way into a site with no parking, or could you see us by video?' And the answer's always been 'No.' So I think some of them are a little bit annoyed because they've been asking for this for years and never received it, and now we've said 'It is possible, after all—who knew?' So, it'll be funny after [COVID-19] to see what people are happy to go back to… My consulting rooms are on the first floor [with no] parking space, which is hideous for older people with mobility problems. So, I think a lot of them would prefer telephone or video rather than having to come in. (Senior clinical psychologist)

## The future of remote care

Service users, carers and staff had varying views on the appropriateness of remote care for secondary mental health services, but generally concurred that shared decision-making about access modes (face to face vs remote care) needed negotiation. As the use of remote care became more common during the pandemic, such consultation and shared decision making became even less common, although largely through force of circumstances. Service users, carers and staff alike felt strongly that organisations should take a tailored approach to introducing remote care once more flexibility returns to the system. They proposed that such an approach should take into account the differences between different types of clinical encounter (eg, psychotherapy, psychiatric reviews, monitoring or changing medication or regular care coordinator encounters) and the range of options for remote contact (including telephone calls, video-supported calls, text messages, emails). They also highlighted that, in the future, mental health services should attempt to acknowledge the unfamiliarity and potential 'strangeness' of accessing services remotely and recognise that people's preferences around remote access may shift in response to their changing circumstances and experiences of mental health:

There wasn't anything about, 'You might find this strange initially; it's going to feel different'… Something like that would have been really nice, like 'It's going to feel different and maybe these are some of the ways that you and your psychiatrist can manage that.' (Service user)

Participants described technology-related and connectivity issues as other key factors to consider in relation to remote access. Wi-Fi and signal problems could affect sound and video quality and sometimes prevented consultations from happening altogether. These issues seemed to be more prominent in rural areas, thus disproportionately affecting services in specific geographical locations.

I know a lot of people who've had technical and/or Information Technology (IT)-related issues with [remote care]—largely due to connectivity or lack of. So that seems to have been the biggest barrier. (Trainee psychiatrist)

There was one appointment when we tried and tried to get the AttendAnywhere to work and it just wouldn't, and it was really frustrating because we could see each other but either I could hear my psychologist, or he could hear me, but we couldn't hear each other. So, we gave up in the end… It does rely on you having a good internet connection. (Service user)

When staff were asked about the future of remote care, most saw it as having a role in supplementing face-to-face contact. Its advantages included avoiding unnecessary or burdensome travel, giving healthcare staff more time to maintain regular contact with service users, allowing the flexibility to offer shorter, more frequent sessions, and perhaps enhanced ability to follow up service users who did not attend their scheduled appointments, although this was not generally the experience of the service users we interviewed. Staff also reported some remote sessions as being more intense than face to face, so they brought them to a close earlier. Finally, staff emphasised that if remote care was going to be more widely used in the future, they would need specific and tailored training for delivering psychological interventions remotely:

Our DNA [did not attend] rate has probably fallen… the standard thing, if somebody doesn't turn up to clinic you just mark that down as did not attend. Although a care coordinator might go out and say to the person… Whereas now what I would do is just phone them up, and I think that's what I will be doing in future. Obviously, you won't always be able to get through, but sometimes you can. And you can learn useful stuff on the phone even if it might not be the same as what you'd see face to face. It's still better than nothing. (Consultant psychiatrist)

I think the telephone consultations have been good. I think it's nice, it offers flexibility to the patients as well as the clinicians. Reduces the need for travel if a

---

**Box 1  Leaning points for remote care**

Learning points and priorities for improvement for providing remote mental healthcare

► While remote consultations allowed secondary mental health services to continue working at a time of unprecedented crisis, refinement is required in how these will be offered in the future.
► The availability of remote access technologies does not mean that direct substitutions for face-to-face care are appropriate.
► Conducting mental health assessments remotely may be particularly problematic and has potential to hinder the identification of risks and use of shared decision making.
► Training for staff in leading or supervising clinical interventions remotely is needed; this must be targeted to the specific remote platform used and be based on co-production principles with service users and carers involved in delivery.
► There is a specific need for guidance on use of remote access mental healthcare, which should be based on further research and consultation.

---

patient had to take time off work, et cetera, to come to an appointment, because before we were quite rigid where we would want them to actually come in. (Trainee psychiatrist)

I was never trained in online therapy… Overnight, you're having to change your practice and it's quite different online… I would always have a piece of paper between me and the client. Well, that was immediately taken away. So, sharing thoughts and formulations became more difficult. (CBT therapist)

## DISCUSSION

This qualitative study of the accounts of 65 service users, carers and healthcare staff of their experiences of remote care during the COVID-19 pandemic offers much rich learning (box 1), including indications of how to optimise service provision in the future and where evidence and guidance is needed. Participants reported both advantages and disadvantages to remote care as a means of sustaining capacity and enabling access to secondary mental health services. Some participants, including both service users and service providers, valued the flexibility offered by remote care, particularly in the context of reduced access to face-to-face service provision.[22] Nevertheless, there were significant downsides. Participants found that consultations by telephone and video restricted therapeutic relationships compared with in-person contact, particularly where service users and staff could not build on a bond already formed face to face. This finding underscores the limitations of the current evidence on video-based consultations as a substitute for in-person healthcare; as Greenhalgh *et al* note, the current literature relates almost exclusively to 'highly selected samples of hospital outpatients with chronic, stable conditions.'[23] Our study further challenges the transferability of this literature to the mental health

context, particularly for service users whose difficulties are fluctuating or who may find themselves in crisis.

Our study has strengths and weaknesses. As a qualitative study, it relies on accounts of behaviours, practices, experiences and opinions as reported by participants. It cannot take into account the clinical or personal outcomes of remote care, or detect causal relationships between these and the various features of remote care identified. Among the study's strengths are its large and varied sample and its novelty in exploring remote care for mental health during a pandemic from the perspective of service users, service providers and family carers. Further strengths include the study's involvement of experts by experience and use of peer research methods, which arguably facilitated more authentic understanding of people's views and experiences, valuing the expertise of all those involved while equally valuing difference. However, some important groups we not included in our study, including people attempting to access mental healthcare for the first time and some key professional groups, including those who work in social care and primary care. The methods of online recruitment and engagement used in the study will have created some barriers for some groups; the approach favoured those to whom we could reach out with information about the study, as well as those with the necessary resource and capacity to decide whether or not to take part in the interview and to complete the informed consent process. For this reason, our findings may underplay the problems. It is possible that inequalities in access to technology may have influenced how participants were able to take part in this study, with 40% of staff taking part in an interview using video methods compared with 20.8% of service users and 20% of carers. Technical difficulties, personal preferences and concerns about confidentiality may account for some differences too. On the other hand, remote methods may also have facilitated involvement of some people who would not have chosen to take part in a face-to-face study.

The study is helpful in identifying the distinctiveness of the mental health context compared with remote care for physical health conditions. Staff and service users alike noted that many features of a consultation that are taken for granted in face-to-face care become problematic in remote consultations. For clinicians, the essential non-verbal cues that are important to their questions, assessments and advice were missing. They sometimes doubted whether service users were willing or able to disclose all relevant information. Similarly, service users and carers felt that important aspects of consultations could easily be missed or misconstrued, especially by telephone, and even during video consultations. Both service providers and service users lamented the loss of the 'sanctity of the consulting room', as a space reserved for highly personal, confidential conversations. Neither the psychological nor the physical features of this space could be replicated in remote consultations. The lack of boundaries between domestic life and the clinical encounter could be immensely stressful, and challenges around privacy

that have been noted in relation to physical health may be particularly difficult in remote care for mental health.[24 25] For some service users, aspects relating to their mental health magnified the challenges that have been noted in remote consultations about physical health.[22 26] Sensory difficulties, communication impairments, digital exclusion and aversion to screen-mediated contact made remote care especially problematic for some service users, and might be caused or exacerbated by some mental health difficulties.

Though remote care in secondary mental health services is likely to become increasingly common, perhaps routine, our findings have important implications for policy, clinical practice and the future development of research in this area. In particular, our work makes clear that a 'one-size-fits all' approach is not a suitable long-term solution once the exigencies of the COVID-19 pandemic have passed. While enhancing access for some groups, remote care may impede it for others, and the differential impact of remote care requires careful evaluation, accounting for impacts on those who withdraw from remote care as well those who engage. This also points to the need to develop tailored, personalised approaches to remote care that cater for the preferences and needs of individual service users, as well as for changes in their mental well-being. Our data show that many service users—and indeed professionals—had limited influence or choice about how care was provided. Identifying the appropriate balance of in-person and remote support for individuals and for different service user groups requires the input of front-line clinicians, service users and families who may be involved in their care, and the adoption of rigorous coproduction methodologies that will take careful development.

Training and development might focus on enabling staff, service users and carers to make the most of the advantages offered by remote care, while identifying and mitigating its challenges. Professional development for staff must evolve, accordingly, as must novel approaches to supporting service users who, presently, are given little information if any as to how best to make the most of remote consultations. While curricula (or a joint curriculum for staff and service users) require further research and pedagogical input, we expect key components to include confidence in use of IT platforms and the means to practise communication skills in virtual settings, consideration of age and cultural contexts, and legal and governance requirements. Equally important for staff is to consider the risks of remote working for members: reduced contact with colleagues in the structured environment of a shared working space may hinder the kind of informal knowledge sharing and mutual monitoring that is key to maintaining safety in healthcare teams. If we can define what good looks like, shared decisions about the option of remote approaches could support a tailored, personal approach to mental healthcare.

In conclusion, the widespread and unavoidable pivoting from face to face to remote mental healthcare during the pandemic was an unplanned natural experiment. It is clear from our work is that guidance is needed on exactly which type of clinical consultations best suit in-person presence, and for whom and why, and which can be offered remotely, and through which medium, taking into account intersectional challenges regarding access that contribute to the continuing 'digital divide' in mental health.[27 28]

**Acknowledgements** We are very grateful to the service users, carers and staff who so generously took part in this study, and those who helped us to develop this study. We would like to thank colleagues at THIS Institute for developing and managing the Thiscovery online research platform that facilitated this work, to Joann Leeding for support with public and patient involvement, to Bethan Everson for assistance with governance and data management, and to Kim Cannon-Sell for help with reference management.

**Contributors** EL was lead researcher and managed the project with NR. PJ, GM, MD-W, JP and VP contributed to the design of the study, analysis and writing-up. EL, NR, JW and JP conducted the interviews. DS, JW and NB contributed to the analysis and writing-up.

**Funding** This research was funded by The Healthcare Improvement Studies Institute (THIS Institute), University of Cambridge. THIS Institute is supported by the Health Foundation, an independent charity committed to bringing about better health and healthcare for people in the UK. Grant number not applicable. All contracted parties contributed to the study under agreements through the same funding. PJ is supported by the NIHR Applied Research Collaboration East of England and by RP-PG-0161–20003. MD-W is an NIHR Senior Investigator (NF-SI-0617–10026).

**Disclaimer** The views expressed in this article are those of the authors and not necessarily those of the NHS, the NIHR, or the Department of Health and Social Care.

**Competing interests** None declared.

**Patient consent for publication** Not required.

**Ethics approval** Ethical approval for the study was obtained from the University of Cambridge Psychology Research Ethics Committee on 15 June 2020, reference: PRE.2020.075.

**Provenance and peer review** Not commissioned; externally peer reviewed.

**Data availability statement** All data relevant to the study are included in the article or uploaded as online supplemental information.

**ORCID iDs**
Elisa Liberati http://orcid.org/0000-0003-4981-1210
Natalie Richards http://orcid.org/0000-0001-5673-751X
Jennie Parker http://orcid.org/0000-0001-5179-729X
Janet Willars http://orcid.org/0000-0002-7886-3223
David Scott http://orcid.org/0000-0001-9083-580X
Nicola Boydell http://orcid.org/0000-0002-2260-8020
Vanessa Pinfold http://orcid.org/0000-0003-3007-8805
Graham Martin http://orcid.org/0000-0003-1979-7577
Mary Dixon-Woods http://orcid.org/0000-0002-5915-0041
Peter Jones http://orcid.org/0000-0002-0387-880X

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
