## [Reviewer comments · BMJ Open]

ARTICLE DETAILS

TITLE (PROVISIONAL)	Remote care for mental health: qualitative study with service users, carers and staff during the COVID-19 pandemic
AUTHORS	Liberati, Elisa; Richards, Natalie; Parker, Jennie; Willars, Janet; Scott, David; Boydell, Nicola; Pinfold, Vanessa; Martin, Graham; Dixon-Woods, Mary; Jones, Peter

VERSION 1 – REVIEW

REVIEWER	Rashmi Patel King's College London Institute of Psychiatry Psychology and Neuroscience, Department of Psychosis Studies
REVIEW RETURNED	05-Feb-2021

GENERAL COMMENTS	Thank you for inviting me to review this manuscript which describes a qualitative study to assess the impact of remote care for mental health during the COVID-19 pandemic. This is an important and timely piece of research given that remote care technology is likely to continue to be used in mental healthcare services beyond the current pandemic. It is particularly important to evaluate the impact of remote care through qualitative studies as its impacts are likely to vary between individuals and healthcare services in a way which cannot be easily captured through quantitative data. It is good that the authors co-produced their work with service users and carers who are the key beneficiaries of this work, and that views were obtained from both patients and carers and the clinicians providing care to these groups. The study is conducted to a high standard in accordance with Standards for Reporting Qualitative Research. The main area I feel the manuscript would benefit from is more discussion of what we should be doing differently in clinical practice and future research in light of the findings from this study. Introduction Clearly and concisely written in a way which is accessible to a global readership. The preceding literature is well summarised, the gaps in our understanding, and how this study fills them. Methods This is a personal point and I will leave it to the authors to decide if they would like to update the wording, but the term “people with mental health disorders” or “mental disorders” may be better than “difficulties” to describe people who access secondary mental health services, in the same way that people accessing acute hospital care suffer with physical health disorders.
---

	It is excellent to see that the NHS staff included in the study were from a range of professional backgrounds. One group which wasn't included was social care and it would be worth highlighting the importance of including social care professionals in future research studies on remote care. It was interesting to see the breakdown of modality for the interviews by telephone vs video call. I wonder if the authors could comment on the significance of a greater proportion of staff taking part in the study by video call rather than telephone and whether this could, in turn, be reflected in barriers towards telephone vs video call in remote mental healthcare (e.g. were staff were better equipped to participate via video than patients and carers and if so, how could this be improved)? It is good that a purposive sampling strategy was used to balance diversity in participants. It would be useful to learn more about the basis to the purposive sampling strategy and what methods were used to assess the ideal balance of participants (e.g. population statistics or mental healthcare service statistics on demographics) and if this is reflected in the breakdown of participants who eventually completed the study. The study was conducted in a robust way with respect to ethics and data security. Results It is interesting to see how patterns of remote care modality have changed during the early phase of the pandemic, with an increasing move to video calling. I wonder if this was reflected in the qualitative data with respect to changes in ease/acceptability of video calling during 2020 and the relative advantages/disadvantages of telephone vs video call? It is helpful to point out the difference between remote care for a completely new interaction compared to an interaction where service user/carer and healthcare professional had previously met in-person. It is also very helpful to learn about modalities of psychological therapy which were less well suited to remote consultation due to the importance of the physical space. The concerns raised by participants with respect to inequities in access are particularly important. We have previously found that rates of remote consultation were much lower for older adults compared to working age adults and children and adolescents (https://www.medrxiv.org/content/10.1101/2020.10.26.20219576v1). Nonetheless, it is heartening to discover that remote care can also be a great enabler for people who have physical impairments which hinder access to in-person care. Discussion The authors do a great job in discussing their findings in the context of previous research. However, what would really add value to the work is to provide more direction to readers in how to respond to the issues raised in their qualitative data analysis. For example, I found it particularly helpful to read about the recommendation of acknowledging that remote consultation is
--	--

	“strange” compared to traditional in-person consultation. It would be great to develop guidelines/training for clinicians which includes this point. The authors point out risks for staff in terms of professional isolation in remote care. There are also risks in terms of fatigue and eye strain. Guidelines should consider practical advice on how a clinician could manage their remote work schedule to prevent work-related health problems. If inequity of access is a problem, what should healthcare service managers/policy makers do to address this? Should we call for the NHS to fund equipment/training for service users/carers/staff? The impact of remote technology on mental healthcare is such an important area during the pandemic and the authors have a tremendous opportunity to help inform policy makers on how best to approach this. In light of this, I wonder if it would be worth expanding upon Box 1 and include a summary (as separate sections within box 1) on recommendations for (i) clinicians, (ii) healthcare services, and (iii) future research? I think this would greatly enhance the impact of the manuscript. It would also help the reader if the authors could complete the article with a short paragraph (a few sentences) of conclusions of the main “take-home” messages.
--	--

REVIEWER	Md. Rabiul Islam University of Asia Pacific
REVIEW RETURNED	11-Feb-2021

GENERAL COMMENTS	Thanks for proving the opportunity to evaluate this paper. I have read the whole paper with great interest. This paper has potential and interesting. However, there are some issues. In the current form, even though the results are of interest to me, I do not find it suitable for publication. Strengths and limitations of this study: Page 3 Line 58: “Strengths include its qualitative approach in speaking to a large sample of participants with varied mental health difficulties, carers, and a diverse range of mental healthcare staff”.  This study included only 65 participants which needs to be justified by appropriate sampling techniques in methods section. Introduction This section appropriately written to justify the need and scope of this study. Methods >>The sampling is critical in evaluating this study. Were the participants from different groups any way representative of the respective population of interest? Please describe the detail sampling technique. What proportion of people approached said no and what was the recruitment rate, that need to be mentioned. Page 7, line 20: You mentioned the semi-structured interviews conducted in the present study. >> what types of questionnaire was applied to collect the responses? Please provide the detail questionnaire used in this study as supplementary data for better understanding of readers. Results The first paragraph does not relate to your study results and are too general to describe the methodology of interviewing. The whole result section describes the qualitative statements/experience of the study participants. Is it possible to perform some statistical analyses in a qualitative study to support your statements? Other comments:
--

	Despite the great efforts of the authors, the data obtained in this project can be published in different format. In my opinion, they are not suitable for full length original research.
--	---

REVIEWER	Christian Dalton-Locke UCL, Division of Psychiatry
REVIEW RETURNED	15-Feb-2021

GENERAL COMMENTS	Intro  - On a couple of occasions, potentially relevant information is referred to in passing and not expanded on: - What were the most important findings from the 42 studies on telehealth in mental health care settings found by Doraiswamy et al ? - What are the “important downsides” referred to in line 53, p.3? Methods:  - Line 4 p.6. I think this is the first time THIS Institute is mentioned – please can “THIS” be spelled out here? - Is it possible to include the topic guide as a supplementary material, and could more detail be provided about the themes explored during interviews (lines 22-25 p.7)? Were these different between the participant groups? Results  - Add column headings to Table 2 Discussion  - Lines 14-16 p.20 – on service user and clinician lack of choice over how care was delivered, can it be added that this was largely, if not entirely, directed by government lockdown measures? I agree with the authors that going forward it is important in mental health care that the right balance between face-to-face and remote care is sought, and tailored to individuals and circumstances, but this balance can only really be struck whilst the option of face-to-face is viable. So, this balance is important to consider moving beyond lockdown but what can be done during lockdown also needs consideration.
--

VERSION 1 – AUTHOR RESPONSE

Proposed responses to the remote care for mental health manuscript, v1.		
ID	Comments to the authors	Authors' response
	Reviewer 1	
R1-1	Thank you for inviting me to review this manuscript which describes a qualitative study to assess the impact of remote care for mental health during the COVID-19 pandemic. This is an important and timely piece of research given that remote care technology is likely to continue to be used in mental healthcare services beyond the current pandemic. It is particularly important to evaluate the impact of remote care through qualitative studies as its impacts are likely to vary between individuals and healthcare services in a way which cannot be easily captured through quantitative data.	Thank you. We are very pleased at this positive response to our manuscript.
R1-	It is good that the authors co-produced their work with service users	We have offered further

2	and carers who are the key beneficiaries of this work, and that views were obtained from both patients and carers and the clinicians providing care to these groups. The study is conducted to a high standard in accordance with Standards for Reporting Qualitative Research. The main area I feel the manuscript would benefit from is more discussion of what we should be doing differently in clinical practice and future research in light of the findings from this study.	discussion (in the Discussion section) on how clinical practice might be informed by the findings of our study and, in particular, the need to develop training and professional development. While the details need to be informed by further research (that we are currently doing) and appropriate pedagogical expertise, we have outlined what we consider to be likely core components.
R1-3	Introduction: Clearly and concisely written in a way which is accessible to a global readership. The preceding literature is well summarised, the gaps in our understanding, and how this study fills them.	Thank you.
R1-4	This is a personal point and I will leave it to the authors to decide if they would like to update the wording, but the term “people with mental health disorders” or “mental disorders” may be better than “difficulties” to describe people who access secondary mental health services, in the same way that people accessing acute hospital care suffer with physical health disorders.	We have considered this suggestion carefully with our collaborators, who include mental health service users. We prefer to retain the term “mental health difficulties”, as this is seen as most acceptable and non-discriminatory in the community being described.
R1-5	It is excellent to see that the NHS staff included in the study were from a range of professional backgrounds. One group which wasn't included was social care and it would be worth highlighting the importance of including social care professionals in future research studies on remote care.	We have now noted the value of including social care professionals in future research studies on remote care.
R1-6	It was interesting to see the breakdown of modality for the interviews by telephone vs video call. I wonder if the authors could comment on the significance of a greater proportion of staff taking part in the study by video call rather than telephone and whether this could, in turn, be reflected in barriers towards telephone vs video call in remote mental healthcare (e.g. were staff were better equipped to participate via video than patients and carers and if so, how could this be improved)?	Thank you, we have reflected on this in the discussion.
R1-	It is good that a purposive sampling strategy was used to balance	We have explained all

7	diversity in participants. It would be useful to learn more about the basis to the purposive sampling strategy and what methods were used to assess the ideal balance of participants (e.g. population statistics or mental healthcare service statistics on demographics) and if this is reflected in the breakdown of participants who eventually completed the study	the factors we considered when inviting participants to take part in an interview. We have also explained that we did not aim to achieve statistical representation of mental health service demographics.
R1-8	The study was conducted in a robust way with respect to ethics and data security.	Thank you.
R1-9	It is interesting to see how patterns of remote care modality have changed during the early phase of the pandemic, with an increasing move to video calling. I wonder if this was reflected in the qualitative data with respect to changes in ease/acceptability of video calling during 2020 and the relative advantages/disadvantages of telephone vs video call?	We have now added a subsection in the 'service user choices about remote care' section to expand on this. We have explained that service users and staff wanted more choice about which remote care modality should be used.
R1-10	It is helpful to point out the difference between remote care for a completely new interaction compared to an interaction where service user/carer and healthcare professional had previously met in-person. It is also very helpful to learn about modalities of psychological therapy which were less well suited to remote consultation due to the importance of the physical space.	Thank you.
R1-11	The concerns raised by participants with respect to inequities in access are particularly important. We have previously found that rates of remote consultation were much lower for older adults compared to working age adults and children and adolescents (https://www.medrxiv.org/content/10.1101/2020.10.26.20219576v1). Nonetheless, it is heartening to discover that remote care can also be a great enabler for people who have physical impairments which hinder access to in-person care.	Thank you.
R1-12	The authors do a great job in discussing their findings in the context of previous research. However, what would really add value to the work is to provide more direction to readers in how to respond to the issues raised in their qualitative data analysis. For example, I found it particularly helpful to read about the recommendation of acknowledging that remote consultation is "strange" compared to traditional in-person consultation. It would be great to develop guidelines/training for clinicians which includes this point. The authors point out risks for staff in terms of professional isolation in remote care. There are also risks in terms of fatigue and eye strain. Guidelines should consider practical advice on how a clinician could manage their remote work schedule to prevent work-	We agree it would be useful to develop guidance and have now made this more prominent in the discussion. We have incorporated several of the useful points made by the reviewer, particularly our response to R1-2, above.

	related health problems. If inequity of access is a problem, what should healthcare service managers/policy makers do to address this? Should we call for the NHS to fund equipment/training for service users/carers/staff? The impact of remote technology on mental healthcare is such an important area during the pandemic and the authors have a tremendous opportunity to help inform policy makers on how best to approach this. In light of this, I wonder if it would be worth expanding upon Box 1 and include a summary (as separate sections within box 1) on recommendations for (i) clinicians, (ii) healthcare services, and (iii) future research? I think this would greatly enhance the impact of the manuscript. It would also help the reader if the authors could complete the article with a short paragraph (a few sentences) of conclusions of the main “take-home” messages.	Thank you. We have fashioned a concluding paragraph, as suggested. We have not amended Box 1 because, on reflection, we felt it worked better as a shorter, overall summary.
	Reviewer 2	
R2-1	Thanks for providing the opportunity to evaluate this paper. I have read the whole paper with great interest. This paper has potential and interesting. However, there are some issues. In the current form, even though the results are of interest to me, I do not find it suitable for publication.	Thank you for the positive comments on the paper. We hope the revisions we have made now render it suitable for publication.
R2-2	Strengths and limitations of this study: Page 3 Line 58: “Strengths include its qualitative approach in speaking to a large sample of participants with varied mental health difficulties, carers, and a diverse range of mental healthcare staff”.  This study included only 65 participants which needs to be justified by appropriate sampling techniques in methods section.	The study, involving 65 participants in a highly sensitive area, is in fact a large sample size by qualitative standards, and is considerably larger than many qualitative studies published by BMJ Open. We have now substantially revised the methods section to offer more clarity, including further detail on the sampling techniques.

R2-3	Introduction: This section appropriately written to justify the need and scope of this study. Methods	Thank you.
R2-4	Methods: The sampling is critical in evaluating this study. Were the participants from different groups any way representative of the respective population of interest? Please describe the detail sampling technique. What proportion of people approached said no and what was the recruitment rate, that need to be mentioned.	We have provided more information about the sampling technique and included data about how many people expressed interest in the study, and of those, how many we invited to take part in an interview (table 1)
R2-5	Page 7, line 20: You mentioned the semi-structured interviews conducted in the present study. >> What types of questionnaire was applied to collect the responses? Please provide the detail questionnaire used in this study as supplementary data for better understanding of readers.	We have now included the interview schedules used in the study as supplementary materials.
R2-6	Results: The first paragraph does not relate to your study results and are too general to describe the methodology of interviewing. The whole result section describes the qualitative statements/experience of the study participants. Is it possible to perform some statistical analyses in a qualitative study to support your statements?	The first paragraph of the results has been amended. We have described the qualitative interview methodology in the Methods section. It not appropriate to perform statistical analyses on these data. We have undertaken high quality qualitative analysis of our qualitative data.
R2-7	Despite the great efforts of the authors, the data obtained in this project can be published in different format. In my opinion, they are not suitable for full-length original research.	This paper reports a well-designed and well-executed large qualitative study, and therefore counts as original research according to BMJ Open's criteria. We hope the revisions to the manuscript have addressed the reviewer's concerns.
Reviewer 3		

R3-1	Introduction: On a couple of occasions, potentially relevant information is referred to in passing and not expanded on:  - What were the most important findings from the 42 studies on telehealth in mental health care settings found by Doraiswamy et al? - What are the “important downsides” referred to in line 53, p.3? 	Doraiswamy et al report a rapid scoping review that quantified the main characteristics of the 543 papers they identified. The review does not provide a summary of the findings of the included studies. Our purpose in referencing it here was to quantify the relative dearth of studies relating to remote mental healthcare compared with other areas of healthcare.
R3-2	Methods: Line 4 p.6. I think this is the first time THIS Institute is mentioned – please can “THIS” be spelled out here?	Done, thank you
R3-3	Methods: Is it possible to include the topic guide as a supplementary material, and could more detail be provided about the themes explored during interviews (lines 22-25 p.7)? Were these different between the participant groups?	Yes, we have now included the interview schedules
R3-4	Methods: Add column headings to Table 2	Done, thank you.
R3-5	Discussion: Lines 14-16 p.20 – on service user and clinician lack of choice over how care was delivered, can it be added that this was largely, if not entirely, directed by government lockdown measures? I agree with the authors that going forward it is important in mental health care that the right balance between face-to-face and remote care is sought, and tailored to individuals and circumstances, but this balance can only really be struck whilst the option of face-to-face is viable. So, this balance is important to consider moving beyond lockdown but what can be done during lockdown also needs consideration.	Thank you, we have addressed this point in the Discussion.

VERSION 2 – REVIEW

REVIEWER	Rashmi Patel King's College London Institute of Psychiatry Psychology and Neuroscience, Department of Psychosis Studies
REVIEW RETURNED	17-Mar-2021

GENERAL COMMENTS	Thank you for inviting me to review the updated manuscript which has fully addressed my comments. It will be a great addition to the literature on remote mental healthcare.
--

REVIEWER	Md. Rabiul Islam University of Asia Pacific
-----------------	--

REVIEW RETURNED	27-Mar-2021
-------------

GENERAL COMMENTS	Thank you for the revision. No further comments on this manuscript.
---